# A subpopulation of cortical VIP-expressing interneurons with highly dynamic spines

Christina Georgiou[1,2], Vassilis Kehayas[1,4], Kok Sin Lee [1,2], Federico Brandalise[1,5], Daniela A. Sahlender [3], Jerome Blanc[3], Graham Knott [3] & Anthony Holtmaat [1✉]

Structural synaptic plasticity may underlie experience and learning-dependent changes in cortical circuits. In contrast to excitatory pyramidal neurons, insight into the structural plasticity of inhibitory neurons remains limited. Interneurons are divided into various subclasses, each with specialized functions in cortical circuits. Further knowledge of subclass-specific structural plasticity of interneurons is crucial to gaining a complete mechanistic understanding of their contribution to cortical plasticity overall. Here, we describe a subpopulation of superficial cortical multipolar interneurons expressing vasoactive intestinal peptide (VIP) with high spine densities on their dendrites located in layer (L) 1, and with the electrophysiological characteristics of bursting cells. Using longitudinal imaging in vivo, we found that the majority of the spines are highly dynamic, displaying lifetimes considerably shorter than that of spines on pyramidal neurons. Using correlative light and electron microscopy, we confirmed that these VIP spines are sites of excitatory synaptic contacts, and are morphologically distinct from other spines in L1.

[1] Department of Basic Neurosciences and the Center for Neuroscience, Faculty of Medicine, University of Geneva, Geneva, Switzerland. [2] The Lemanic Neuroscience Graduate School, Universities of Geneva and Lausanne, Geneva, Switzerland. [3] Ecole Polytechnique Federale Lausanne, Lausanne, Switzerland. [4] Present address: Institute of Computer Science, Foundation for Research and Technology  Hellas (FORTH), Heraklion, Crete, Greece. [5] Present address: Department of Bioscience, University of Milan, Milan, Italy. ✉email: anthony.holtmaat@unige.ch

Synaptic changes including the formation and elimination of synapses may underlie neuronal network plasticity. The majority of excitatory synapses in the neocortex appear on spines[1]. Almost all spines contain a synapse[1–3], and spine size positively correlates with synapse size[4–6] and synaptic currents[7,8]. Therefore, morphological changes of spines can be interpreted as synaptic changes[9,10]. Longitudinal imaging studies have revealed that changes in spine morphology or composition are ongoing phenomena that occur throughout an animal's life[11–14], and that spine formation and elimination on pyramidal neurons is correlated with changes in experience and learning[9,11,15–21]. Similar to pyramidal neurons, spines on inhibitory neurons are sites of excitatory synapses[22–25]. Dendrites and spines of some interneurons display morphological changes under baseline conditions and upon sensory deprivation[23,26–28]. However, due to the great variety in interneuron subtypes, insight into morphological plasticity of inhibitory interneurons remains incomplete.

In the cortex the three most prevalent non-overlapping inhibitory neurons are parvalbumin (PV), somatostatin (SST), and serotonergic receptor 3a (5HT3aR)-expressing neurons[29,30]. The latter is divided into two subgroups depending on the expression of VIP. Apart from providing inhibitory inputs to pyramidal neurons and granule excitatory neurons, many inhibitory neurons are also interconnected, thereby forming disinhibitory circuits[31]. Disinhibitory circuits provide a key motif for associative learning[32,33]. VIP-expressing neurons constitute an important component of cortical disinhibitory circuits. When activated they inhibit mainly SST or PV interneurons, which in turn inhibit pyramidal neurons[31,34]. Some VIP interneurons that are located in upper cortical layers may receive long-range excitatory and modulatory input[35–43], and have spines[22,44]. They play a role in motor integration in various cortical areas[38,45,46], in gain control during sensory discrimination[37], and in cortical plasticity[41,47].

Here, we tracked synaptic changes on VIP interneurons in vivo. We used a VIP-specific Cre driver line[48] and Cre-dependent adeno-associated viral (AAV) vectors to express recombinant enhanced green enhanced fluorescent protein (GFP) in VIP interneurons. Using 2-photon laser scanning microscope (2PLSM) in vivo we demonstrate that a subset of multipolar VIP neurons, located in superficial cortical layers, have high spine densities on their dendrites that ramify in cortical L1. Their morphological and electrophysiological characteristics are similar to the bursting type of small VIP cells[22,44,49–52]. Long-term imaging, over days to weeks, revealed that their spines have strikingly different dynamics as compared to spines of pyramidal neurons, displaying a higher instantaneous probability of disappearing. In addition, we provide evidence that VIP-cell spines show considerable variance in their probability of survival. Using correlative 2PLSM and serial section scanning electron microscopy (SSEM), we show that VIP spines, including newly formed spines, are sites of excitatory connections. Spiny VIP dendrites have synapse densities similar to a control dendritic segment in the same cortical area, but their excitatory synapses and spines are smaller. In addition, single spiny VIP dendritic branches have multiple synaptic connections with single axons (compound connections), and a relatively large fraction of their spines have multiple excitatory contacts. Altogether, our data indicate that spines on a subpopulation of VIP neurons are potential sites of a sustained and distinct type of cortical plasticity.

## Results

**Expression of GFP in spiny, multipolar VIP interneurons, located in superficial layers of the cortex.** In order to target VIP-expressing interneurons for imaging, we used the VIP-IRES-Cre transgenic mouse line[48]. We performed small local injections of a Cre-dependent AAV vector to target GFP expression to VIP neurons located in the superficial layers (L1 and L2/3) of the mouse barrel cortex (Fig. 1a). A cranial window was implanted immediately after the AAV injection, and 2PLSM in vivo was performed at least two weeks later. We observed small clusters of neurons expressing GFP (Fig. 1b) confined to superficial L1-3. L1 abounded in fluorescently labeled spiny dendrites extending from those neurons (Fig. 1c, d), which we could also observe in densely tdTomato-labelled cortex of VIP-IRES-Cre mice crossbred with the Ai14 reporter line[53] (Supplementary Fig. 1). We continued our experiments using the AAV-approach, which provided a sparse expression pattern that was limited to a subset of superficially located neurons (Fig. 1a, b).

Reconstructions of the 3D morphology of the spiny GFP-positive cells ($n = 6$) in 2PLSM z-stacks (Fig. 1d) indicated that they had a multipolar morphology with somata located approximately between L1 and L2/3 (~150 μm below the pia). Typically, 4 or 5 principal dendrites emerged from the soma (4.16 ± 0.95 dendrites/cell), but the morphology of the dendritic arbors varied considerably. Some had short and relatively simple arbors (e.g. 22 branch tips), with the longest branches reaching to approximately 150 μm from the soma. Others had large and complex arbors (e.g. 31 branch tips), which spanned distances up to 250 μm (Supplementary Fig. 2).

To further characterize the GFP-positive cells, we performed post-hoc immunofluorescence for markers of three distinct inhibitory neuronal populations[29,54,55] (but see,[56]) as well as for subpopulations of cortical VIP neurons[30,55]. At first inspection it appeared that many GFP-positive cells were labeled with anti-VIP antibodies, some with anti-calretinin (CR), neuropeptide Y (NPY), or cholecystokinin (CCK) antibodies, but none with anti-SST or anti-PV antibodies (Supplementary Fig. 1 and 3). To quantify this, regions of interest (ROI) around GFP positive cell bodies were defined by algorithmically thresholding the image (1167 ROIs, 62 slices, 8 mice). We estimated the probability of a ROI to be positive for VIP or for a set of other molecular markers using a Bayesian generalized linear model (GLM). We used variables calculated from the fluorescence intensity of antibody staining as predictors, adjusted with variables extracted from the GFP fluorescence profile of each ROI, and used priors based on co-expression rates provided in the literature. A receiver operator characteristic curve (ROC) showed that our model achieved satisfactory performance (Supplementary Fig. 4). The posterior probability estimated for the different molecular markers showed that cells identified based on GFP labeling, therefore likely VIP-positive, were often labeled with anti-VIP (0.70 ± 0.08), sometimes with anti-CR (0.36 ± 0.05), occasionally with anti-NPY or anti-CCK (0.03 ± 0.03, 0.07 ± 0.07, respectively), and never with anti-SST or anti-PV antibodies. This confirms that GFP expression had not 'leaked' into other inhibitory classes, which is in agreement with previous studies[37,48]. The relatively large proportion of co-labeling with anti-CR antibodies agrees with the finding that many VIP neurons in the upper layers of cortex are CR-expressing bipolar cells[52,55,57,58]. The small proportions that were co-labeled with anti-CCK and anti-NPY antibodies is also congruent with previous findings[55–57]. We could identify various GFP-expressing cells with spiny peri-somatic dendrites that were positive for CCK or CR (Supplementary Fig. 5). This was confirmed in 2 mice in which the cell bodies of vivo-imaged dendrites were identified in perfusion-fixed brain sections and immunolabeled (Supplementary Fig. 6). 4 out of 11 imaged dendritic trees (GFP-expressing) were spiny. Of those 1 was co-labeled with CCK and 1 with CR. The other two expressed neither one of these two markers.

Altogether, the data suggest that the superficial spiny GFP-positive neurons are VIP neurons with a multipolar morphology and may be part of a variety of classical VIP subtypes, among

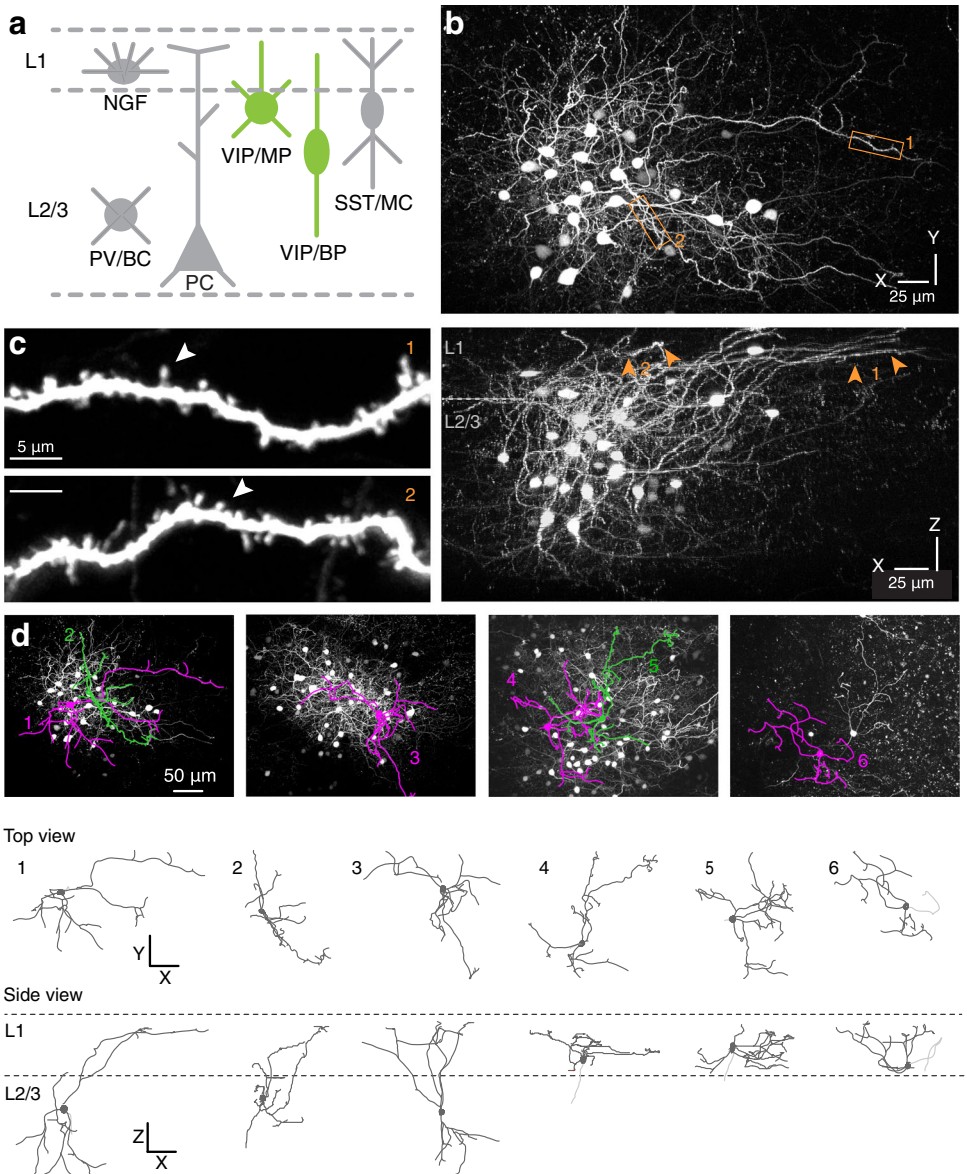

**Fig. 1 Fluorescent labeling and imaging of spiny neurons in the VIP-Cre-driver line. a** Schematic of the different interneuron subclasses in superficial cortical layers. VIP neurons (green) were targeted for labeling. **b** Top: a maximum intensity projection (top view) of a 2PLSM z-stack spanning the entire injection site harboring GFP-expressing VIP neurons. Bottom: side view of the same injection site, indicating that GFP expression is limited to neurons in superficial L1-3. Arrows point to superficial spiny dendrites in L1. **c** High-magnification images of spiny dendritic segments (boxed areas in **b**), with arrowheads pointing to examples of spines. **d** Maximum intensity projections of 2PLSM stacks of GFP-positive VIP neurons and their neurolucida reconstructions. The top and side views show the extent of the dendritic branching patterns and the varying multipolar morphologies of GFP-expressing VIP neurons. Some axons were found to descend towards deeper layers (light grey).

which CCK or CR expressing types. Morphologically, the cells bear similarities to the spiny multipolar VIP cell types that have been previously described in the rat visual cortex[44].

**Spiny VIP neurons have distinct electrophysiological properties.** To further compare the spiny cells with previously classified VIP neurons, we performed somatic patch-clamp recordings in brain slices (Fig. 2). VIP neurons were labeled using a Cre-dependent AAV vector encoding mCherry. For recordings, we targeted superficial (L1/L2) mCherry-expressing neurons (within ~200 μm from the pial surface) with a non-bipolar morphology reflecting the morphologies of the neurons imaged in vivo. During the recordings, cells were filled with biocytin in order to post-hoc classify them as spiny or non-spiny (Fig. 2a). Upon current injections in current-clamp configuration, spiny cells had significantly

larger voltage sags ($3.3 \pm 0.5$ mV, $n = 9$, vs $1.2 \pm 0.4$ mV, $n = 10$; $p = 0.001$) and rebound amplitudes ($3.8 \pm 0.5$ mV, $n = 8$, vs $1.8 \pm 0.6$ mV, $n = 10$; $p = 0.045$), and they displayed pronounced depolarizing after potentials (DAP) ($3.8 \pm 0.5$ mV, $n = 8$), which were not detected in 10 out of 10 non-spiny cells (Fig. 2b, c). Their action potentials had smaller amplitudes ($54.6 \pm 2.2$ mV; $n = 8$, vs $62.0 \pm 2.3$ mV, $n = 10$; $p = 0.036$) (Fig. 2c). In addition, all spiny neurons were characterized by burst firing (duration of the inter-spike intervals ISI: $5.4 \pm 0.8$ ms; $n = 6$, vs $15.1 \pm 4.9$ ms, $n = 8$; $p = 0.018$) (Supplementary Fig. 7). Both spiny and non-spiny neurons were mostly adapting (adaptation index: $0.65 \pm 0.1$; $n = 6$, vs $0.67 \pm 0.08$ ms, $n = 8$; $p = 0.89$)[59]. Furthermore, voltage clamp experiments showed that spiny cells had a larger capacitance ($32.0 \pm 1.4$ pF, $n = 9$, vs $23.3 \pm 1.4$ pF, $n = 10$; $p = 0.002$), and longer time constants upon hyperpolarization ($0.27 \pm 0.02$ ms, $n = 8$, vs

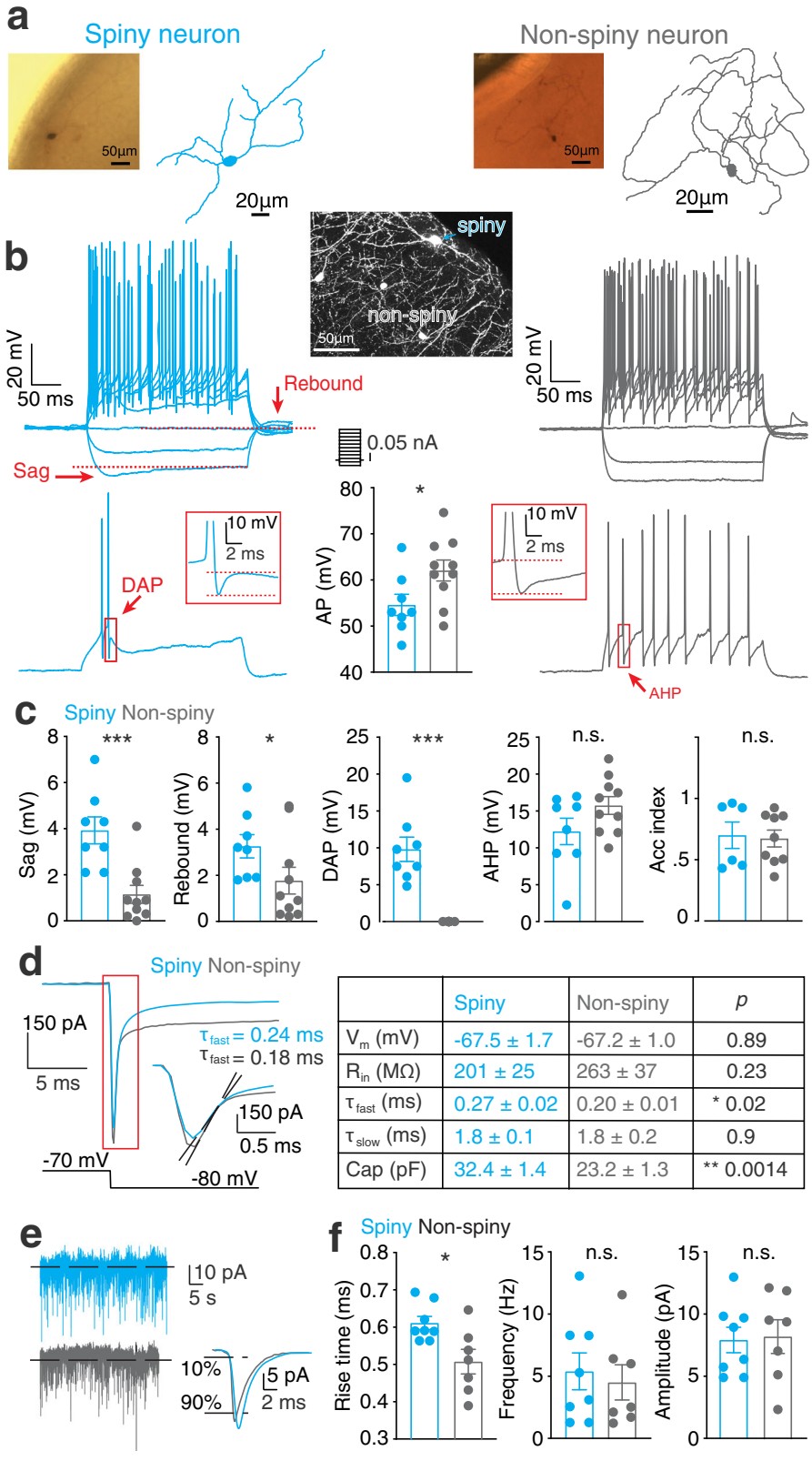

0.20 ± 0.01 ms, $n = 10$; $p = 0.002$) (Fig. 2d). In line with their passive properties, the rise times of the spontaneous excitatory post synaptic currents (sEPSC) (Fig. 2e) were longer (0.60 ± 0.01 ms, $n = 8$, vs 0.50 ± 0.03 ms, $n = 7$; $p = 0.019$) (Fig. 2f), while the amplitudes and frequency of these events were not significantly different (7.9 ± 1.0 pA, $n = 8$, vs 8.2 ± 1.4 pA, $n = 7$; $p = 0.84$). These characteristics align with the electrophysiological signatures of superficially located burst-spiking VIP cell populations[49,50], which may predominantly consist of CCK-expressing neurons[50].

**Dendritic spines on VIP neurons are dynamic**. To study the dynamics of VIP spines we repeatedly imaged 16 dendrites in 5 mice over a maximum period of 54 days with a minimum and

**Fig. 2 Electrophysiological properties of spiny VIP neurons. a** Examples of biocytin staining and corresponding reconstructions of superficial VIP-positive spiny (left) and non-spiny (right) neurons. Inset shows identification of spines (sp; arrow). **b** Top, firing patterns and action potential (AP) properties were measured by current injection steps of increasing amplitude (0.05 nA). In the example, the spiny neuron (left) shows a pronounced sag at the most hyperpolarizing step while no sag was detected for the non-spiny neuron (right). Bottom, the AP properties were analyzed at the first positive current step that triggered APs. Spiny neurons are characterized by the presence of depolarizing after potential (DAP; see inset on the left) while non-spiny neurons do not express this. The action potential amplitude was significantly higher in non-spiny cells. **c** Pooled data of all the recorded neurons for sag, rebound, DAP, after hyperpolarization (AHP), and accommodation (Acc) index. **d** Left, neuronal passive properties as calculated in voltage clamp by a hyperpolarizing step of 10 mV (from -70 to -80 mV). Inset shows representative traces for a capacitance transient, of which the slope (fitted with the black dotted lines between 10 to 90% of the amplitude) is steeper for the non-spiny VIP cell. Right, the analyzed parameters are shown in the table on the right. Note that, consistently with morphological differences, the capacitance and membrane time constant are significantly different between the two groups.
**e** Spontaneous excitatory postsynaptic currents (EPSC) in -70mV voltage clamps. Inset shows examples of averaged spontaneous EPSCs for two cells, indicating a slower rise time for the spiny VIP cell (between 10 to 90% of the amplitude). **f** Average of all the spontaneous events recorded in the traces shown in (e). The rise time of the EPSC for the spiny neurons is significantly longer as compared to the non-spiny cells.

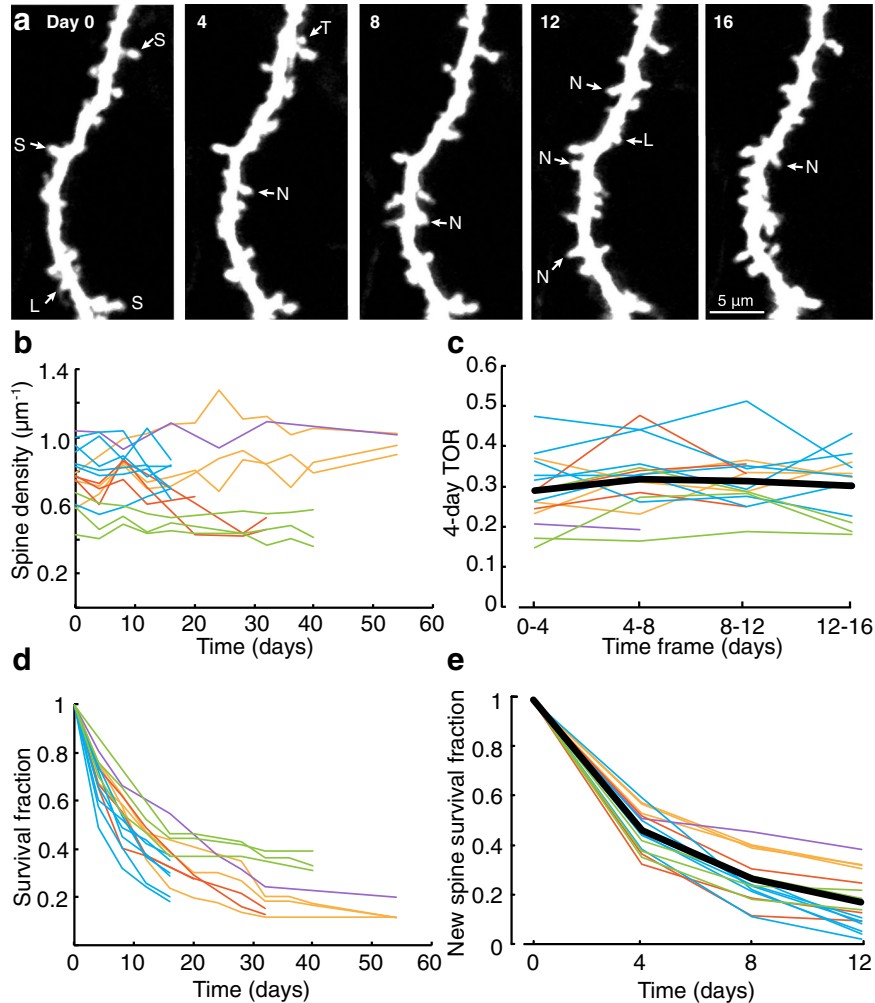

**Fig. 3 Spines on VIP neurons are dynamic. a** Time-lapse images of a VIP-GFP dendritic segment in L1. A number of spine types can be detected, varying from stable (S), new (N), and lost (L) spines. Some new spines are lost immediately (transient, e.g. T) **b** Spine densities over time. Lines correspond to individual dendrites. Lines with the same color belong to the same mouse. **c** The 4-day spine turnover ratio (4-day TOR): the fraction of spines that is gained and lost between two imaging sessions. **d** The survival fractions of spines that were observed on the first day of imaging (day 0). **e** The survival fractions of newly formed spines. Thick black lines in (**c**) and (**e**) represent averages.

maximum interval of 4 and 22 days, respectively (Fig. 3a). Spine density strongly varied between dendrites (from 0.36 µm⁻¹ to 1.27 µm⁻¹), but remained stable on average (0.75 ± 0.17 µm⁻¹) over time ($p = 0.68$, F = 0.17, DF = 103.76, $F$-test on linear mixed-effects model, Fig. 3b). Thus, the spine loss that was observed throughout the imaging period was compensated by

spine gains, which kept the total spine density constant over time. This suggests that the spine dynamics are homeostatically regulated and that the imaging procedures had not detectably affected the physiology of spines.

To quantify the relative spine stability, we calculated the 4-day turnover ratio (TOR) over a maximum of 4 imaging intervals

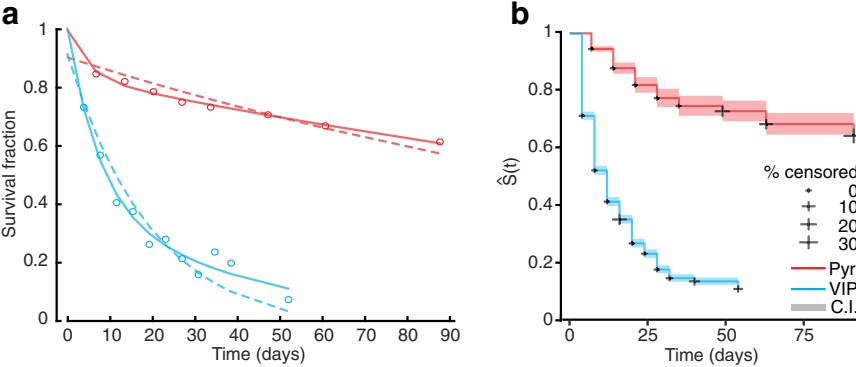

**Fig. 4 Comparison of VIP spine dynamics with L2/3 pyramidal spine dynamics. a** The survival fraction of spines that were present in the first imaging session (day 0) from VIP (blue) and pyramidal neurons (red) were fitted with single ($f(t) = ae^{bt}$, dashed lines) and two-exponential-sum ($f(t) = ae^{bt} + ce^{dt}$, solid lines) functions. The data for the two neuronal classes (circles) represent the average over mice. **b** The spine survival probabilities plotted as the Kaplan–Meier estimate. Shaded areas represent 95% confidence intervals.

(Fig. 3c). The TORs varied between individual dendrites (ranging between 0.15 to 0.51), but their average ($0.31 \pm 0.05$) remained constant over 4 imaging intervals ($p = 0.18$, F = 1.8, DF = 90, F-test on linear mixed-effects model).

Next, we calculated the survival fraction of spines as a function of time. Of the spines that were observed on the first day of imaging ($n = 948$ spines, $n = 16$ dendrites, $N = 5$ mice, day 0, Fig. 3d), slightly more than half ($0.55 \pm 0.11$) survived over the first 8 days. For the dendrites that could be imaged for 32, 42 or 54 days, survival of the initial population continued to decrease to approximately one tenth (32 days: $0.24 \pm 0.10$, $n = 9$ dendrites; $N = 4$ mice; 40 days: $0.25 \pm 010$, $n = 6$, $N = 2$; 54 days: $0.14 \pm 0.04$, $n = 4$, $N = 2$; Fig. 3d), indicating that spines surviving for the first 8 days can still be vulnerable to loss.

Spines that were newly formed within the imaging period were also unstable. Less than half of the new spines survived over the first 4 days after their appearance ($0.47 \pm 0.08$) and their survival fraction continued to decline (8 days: $0.27 \pm 0.10$; 12 days: $0.18 \pm 0.11$; Fig. 3e). Altogether, the data indicate that the total spine population is largely unstable.

**Dendritic spines on VIP neurons are less stable as compared to pyramidal cell spines.** The homeostatically regulated spine turnover is reminiscent of the structural dynamics that have been observed on L5 and L2/3 cortical pyramidal neurons. However, based on the survival fraction, in the current data set the entire spine population appeared to be rather unstable. In pyramidal neurons the vast majority of spines were found to be stable for several days to weeks[11,12,14,16,19], which was observed as deviations of the spine survival fraction from the exponential decay function[11,12,19]. In order to facilitate comparisons with this previous literature, we fitted the spine survival fraction with a single exponential and a two-exponential-sum function. We found that a two-exponential-sum function accounted for more information in the data, even when considering the increase in the number of parameters, as assessed by the corrected Akaike Information Criterion (cAIC) (Fig. 4a). Consistently with the existing literature, the same was true for spine survival fractions of L2/3 pyramidal neurons taken from a previous study[60] ($n = 237$ spines, $n = 5$ neurons, $N = 3$ mice) (Fig. 4a). Spine survival probabilities from both neuronal classes displayed considerable variance and the same was true for new spines of VIP neurons (Supplementary Fig. 8). However, the two-exponential-sum coefficient estimates differed considerably between VIP and pyramidal neurons

$$S_{vip}(t) = 0.59e^{-0.13t} + 0.42e^{-0.02t} \qquad (1)$$

$$S_{pyr}(t) = 0.16e^{-0.21t} + 0.84e^{-0.00t} \qquad (2)$$

In order to directly compare these two classes of neurons, we fitted the survival data with a Cox proportional hazards model (Fig. 4b). The model does not support the hypothesis that spine survival follows a similar curve between the two neuronal classes and we can estimate that spines of VIP neurons have $\approx 7.02$ times higher instantaneous probability to disappear as compared to spines of pyramidal neurons ($b \cong 1.95 \pm 0.18$, $z \cong 11.15$, $p < 0.001$).

**Dendritic spines on VIP neurons are sites of excitatory synapses.** To investigate subcellular and synaptic morphology of the spiny VIP cells, we performed correlated light and electron microscopy of GFP-expressing dendritic segments[61–63], using two complementary scanning electron microscopy (SEM) techniques (Figs. 5 and 6). Focused ion beam SEM (FIBSEM; Fig. 5) provides a high-resolution view of a small piece of dendrite (Fig. 5)[62,64], while serial block face EM (SBEM) with the 3View system (Fig. 6) allows reconstruction of a larger volume (Fig. 6), but at a slightly lower resolution as compared to FIBSEM[65].

We reconstructed 3 GFP-filled dendritic segments (161 μm in 3View, and 23 μm in FIBSEM) in 2 mice. The time-lapse images of the dendrites allowed us to distinguish stable and new protrusions (Fig. 5a, Fig. 6a–d). The vast majority of the 83 protrusions (74 on the two SBEM dendrites, and 9 on the FIBSEM dendrite) seen on the last day of imaging could be found in the EM reconstructions. Several additional protrusions were found in the EM that were obscured by the dendritic shaft in the LM images. Conversely, some very thin protrusions that had been seen in the LM images could not be found in the EM, possibly due to structural alterations just prior to the perfusion fixation. Of the spines that were visible in the 2PLSM images on the last day and subsequently detected in EM, 13% ($n = 11$) had been newly formed within the time frame of imaging (e.g., #13, #14, #15, #16 in Fig. 5a,b; #10, #21, #25, #27 in Fig. 6c, d). Various atypical spines or complex protruding structures were also found (e.g., #3, #14, #15 in Fig. 5b; asterisks Fig. 6a; and #12 in Fig. 6b).

Most spines, including the new ones (10 out of 11; e.g. S5 on #10 in Fig. 6c'), had at least one asymmetric synapse (Table 1), although some synapses were found only at the spine base and not on the tip (e.g., #3', #13 in Fig. 5b; S3 on #3 in Fig. 6b). This indicates that akin to spines of pyramidal neurons and spines of

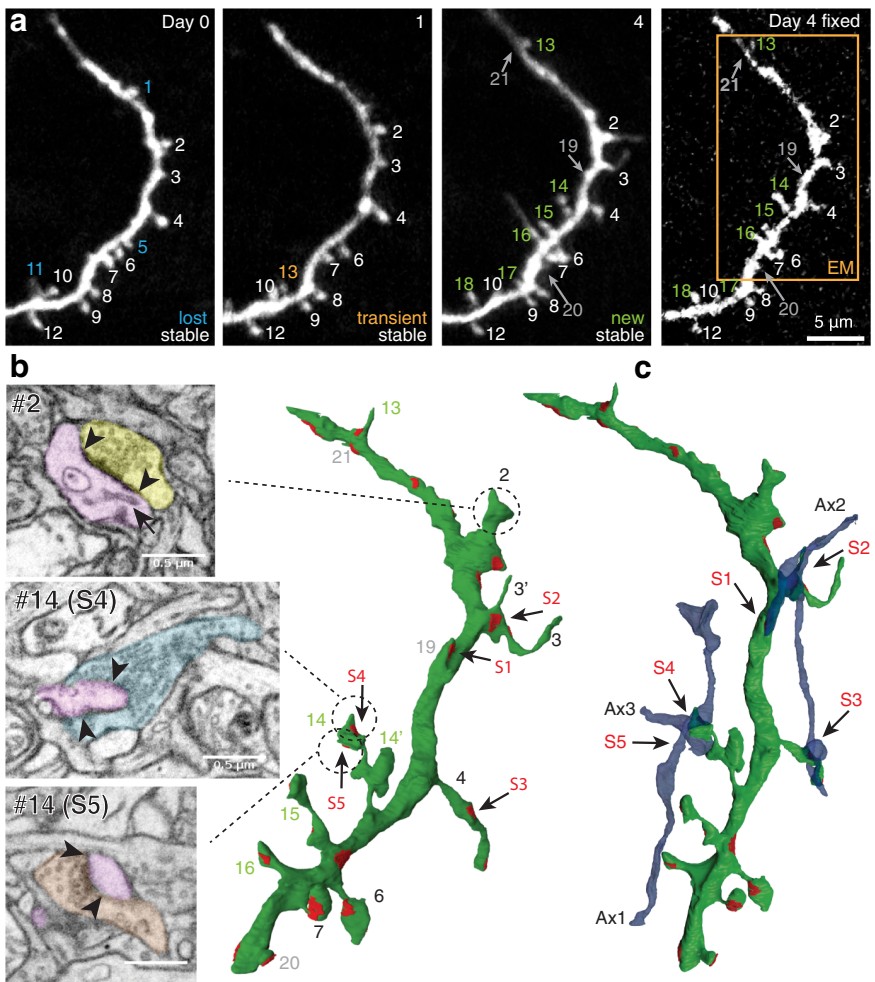

**Fig. 5 Spiny VIP dendrite, imaged in vivo and reconstructed in FIBSEM. a** Time-lapse images of a VIP-GFP dendritic segment that was later reconstructed using FIB/SEM. Spines are tracked and numbered over time. Stable, lost, transient, and new spines could be identified. The image on the far right depicts the same dendritic segment in a vibratome section of the perfusion-fixed brain. All spines that were seen on the last day (day 4) in vivo could be found the fixed material. **b** Middle, the FIBSEM reconstruction (green) of boxed region on the dendritic segment in (**a**). Synapses are indicated in red. All of them were asymmetric. All spines that were seen on the last day of imaging in vivo were recovered in the EM reconstruction. Grey numbers indicate spines that were obscured by the dendritic shaft in the in vivo image. Left, electron micrographs of spines 2 and 14. Presynaptic elements are in yellow, blue and orange; postsynaptic elements in pink. Arrowheads demarcate the synaptic contacts. The arrow in spine 2 indicates ER. Spine 14 contained two separate synapses on its head. Some spines had a compound morphology (e.g., spine 3, 14 and 15) or various thickenings (e.g., spine 2). **c** Reconstructions of three presynaptic axons (blue) showing the compound connectivity of the reconstructed dendrite. Upper panel, an axon that has 3 synaptic contacts in total, 2 with different spines (S1 and S3) and one on the shaft (S2). Lower panel, two different axons that contact the same spine (spine 14, S4 and S5). Note that spine 14 was newly formed.

various other inhibitory neurons[22–24,66], spines on VIP neurons are sites of excitatory synaptic contacts.

To get a sense of the distinctive spine and synaptic features of the VIP dendrites, we also reconstructed a GFP-negative control (CTR) dendrite in the volume of tissue imaged with the SBEM, which had features similar to pyramidal cell dendrites in L1 of S1[2,67] (Fig. 6a). The length and branching structure of the CTR dendrite was comparable to the VIP segments, but its shaft was generally thicker. A quantitative comparison between the two VIP dendrites and CTR dendrites revealed some striking differences and similarities (summarized in Tables 1 and 2, Fig. 6e–g, and Supplementary Fig. 9).

1. The reconstructed VIP dendritic segments had a total of 340 synapses (1.92 μm$^{-1}$), of which 80% had the hallmarks of glutamatergic contacts. They were asymmetric, with a thick postsynaptic density [PSD]), and their presynaptic boutons contained round, electron transparent vesicles[68].

10% were symmetric (presumably GABAergic) and 10% could not be classified. On the CTR dendrite we found a similar density of synapses (1.84 μm$^{-1}$), and a similar percentage of excitatory synapses (79%). The percentage of inhibitory synapses (19%) on the CTR dendrite was somewhat higher, depending on how the unclassified synapses on the VIP dendrite would rank. Nonetheless, this is in line with the general ratio of excitatory and inhibitory synapses in cortex (i.e. 80/20%)[69].

2. Whereas the surface area of the inhibitory synapses was similar (VIP: 0.12 ± 0.07 vs. CTR: 0.13 ± 0.07 μm$^2$), the excitatory synapses were significantly smaller on VIP dendrites as compared to the CTR dendrite (VIP: 0.13 ± 0.08 vs. CTR: 0.19 ± 0.15 μm$^2$, $P < 0.0001$; Fig. 6f, g). The fraction of excitatory synapses that was found on VIP spines was relatively low (27% for VIP vs 75% for CTR), whereas the fraction of inhibitory synapses was relatively high (23% for VIP vs 13% for CTR). Nonetheless, a

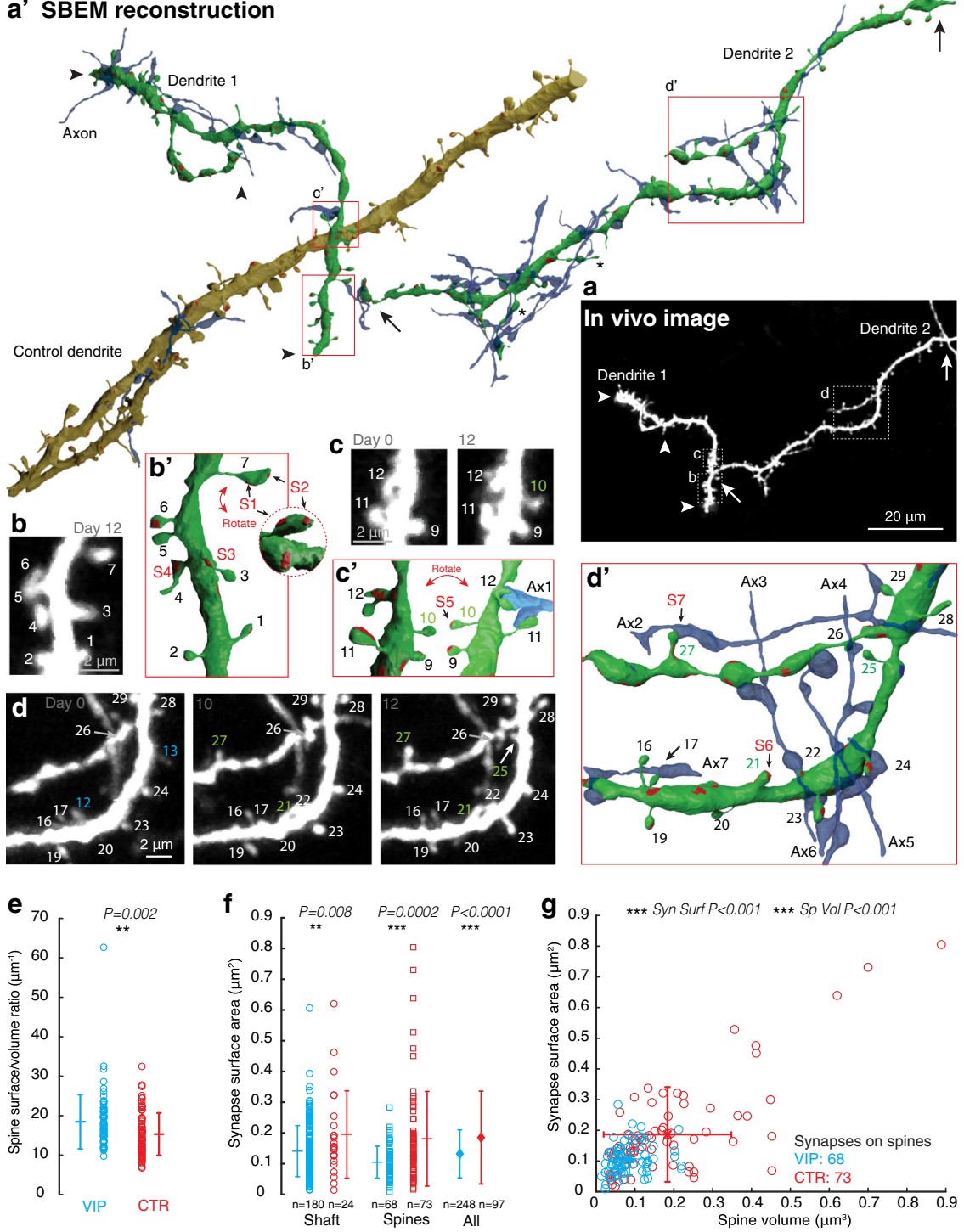

**Fig. 6 Ultrastructural analysis of spiny VIP dendrites, their synapses, and compound presynaptic axons. a** Best projection of in vivo images of two VIP dendrites. Arrows and arrow heads demarcate the dendritic segments that were reconstructed. (**a'**) SBEM reconstructions of the 2 VIP-dendrites (green, dendrite 1, 2) and 1 control dendrite (olive), and their presynaptic axons with compound connections (blue), i.e. more than 1 synapse with the same dendritic segment. Boxes indicate the high-magnification views in (**b**–**d** and **b'**–**d'**). Asymmetric synapses are in red, others in blue. (**b, c, d**) high magnification in vivo images and (**c', d', e'**) detailed reconstructions of dendritic sub-regions. Spines are numbered. Green numbers (#10, #21, #25, and #27) represent newly formed spines. Blue numbers (#12 and #13) are spines that were lost over the time frame of imaging. Axons with multiple connections are in blue and numbered (Ax1, Ax2 etc). **e** The spine surface-to-volume ratio is significantly larger for VIP spines as compared to CTR spines. **f** Excitatory synapse surface of shaft and spine synapses is significantly smaller on VIP dendrites as compared to CTR. **g** The relationship between spine volume and excitatory synapse surface.

**Table 1 Spine data.**

| Type | # Spines [Dens (μm⁻¹)] | Spines with Syn | Spines >1 Syn | Spines >1 ExcSyn | Spines w. Spine App | Vol ± s.d ($μm^3$)* | SA ± s.d. ($μm^2$)* | Exc SynSA/ SpVol | SpExcSyn with MSB |
|---|---|---|---|---|---|---|---|---|---|
| VIP | 77 [0.48] | 56 [73%] | 19 [25%] | 12 [16%] | 61 [79%] | 0.07 ± 0.05 | 1.04 ± 0.49 | 1.54 ± 0.97 | 7 [10%] |
| CTR | 80 [1.20] | 68 [85%] | 7 [9%] | 4 [5%] | 61 [76%] | 0.16 ± 0.16 | 1.88 ± 1.16 | 1.28 ± 0.78 | 6 [8%] |

# number, *Dens* linear density, *Syn* synapse, *ExcSyn* excitatory synapse, *w. Spine App* with spine apparatus, *Vol* mean volume, *SA* mean surface area, *Sp* spine, *MSB* multisynaptic bouton, *s.d.* standard deviation, * significant difference (see Fig. 6).

relatively large fraction of VIP spines contained more than one synapse (25% in VIP vs 9% in CTR) and more than one excitatory synapse (16% in VIP vs 5% CTR; e.g. #14 in Fig. 5b, c; #7 in Fig. 6b'). Interestingly, 4 out of the 9 new spines that could be detected in EM had at least two excitatory synapses (e.g., #14, #15 in Fig. 5b, c; #27 in Fig. 6d').

3. VIP spines had smaller volumes (Vol) and surface areas (SA) as compared to CTR spines (Vol: 0.07 ± 0.05 in VIP vs 0.16 ± 0.16 in CTR, $P < 0.001$; SA: 1.04 ± 0.49 in VIP vs 1.88 ± 1.16 in CTR, $P < 0.001$; Supplementary Fig. 8). Nonetheless, the spine SA to Vol ratio was larger for VIP spines (18.4 ± 6.9 for VIP vs 15.3 ± 5.3 for CTR, $P = 0.002$; Fig. 6e), and the synapse SA to spine Vol ratio was not significantly different (1.54 ± 0.97 for VIP vs 1.28 ± 0.78 for CTR, $P = 0.07$; Supplementary Fig. 9).

4. Endoplasmic reticulum (ER) was found in the majority of both types of spines (79% in VIP vs 76% in CTR), which included the newly formed spines on the VIP dendrites (e.g., #2 in Fig. 5b). In both types, approximately 8–10% of the excitatory synapses on spines were formed with boutons that also contained synapses with other spines (multi-synapse boutons, MSBs). Both of these parameters approximately align with previous observations on pyramidal dendrites in L1 of S1[2].

5. On the VIP dendrites we found that a relatively large fraction (18%) of the synapses were with axons that had at least one additional synaptic contact with the same dendritic segment (e.g., S1, S2 and S3 with Ax2 in Fig. 5c; Ax1 with #12 and #11 in Fig. 6c'; Ax2-7 in Fig. 6d'). This was the case for only 5% of the synapses found on the CTR dendrite. Synapses with such axons have previously been termed 'compound connections'[70] or 'redundant synapses'[71]. In the SBEM imaged block, we estimated that 8.7% of the axons that made a synapse with the VIP dendrite (0.15 μm⁻¹) had such compound or redundant connections, whereas this was the case for only 2.5% of the axons contacting the CTR dendrite (0.05 μm⁻¹). Various configurations of compound contacts were encountered, such as repeated dendritic spine (e.g. Ax1 in Fig. 6c') or shaft synapses (e.g., Ax5 in Fig. 6d'), shaft and spine synapses (e.g. Ax2,3,4,6, and 7 in Fig. 6d'), and synapses on distant spines (e.g., Ax2 in Figs. 5c and 6d').

Altogether, the EM data reveal a marked difference between the VIP and CTR dendritic segments in their spine and synapse phenotypes, as well as their connectivity diagrams. Spines and excitatory synapses are smaller on VIP dendrites. However, VIP dendrites tend to have more spines with multiple excitatory synapses, and tend to be more densely connected to single axons through compound synaptic contacts.

## Discussion

We describe a subpopulation of small multipolar VIP neurons, located in superficial cortical layers and with high dendritic spine densities in L1. The population spans a molecularly diverse subset of VIP neurons, but their electrophysiological characteristics are similar to bursting cells[49]. Their dendrites and spines tend to be more densely connected to axons through compound synaptic contacts than pyramidal dendrites, yet their spines and synapses are generally smaller. The spines are highly dynamic, with appearance and disappearance rates that exceed previous reports of cortical pyramidal neurons, and of other inhibitory neurons. Yet, these new spines make synaptic contacts. Thus, our study identifies structural plasticity of VIP inhibitory neuronal subtypes that have thus far remained poorly characterized. It also implies that spine structural plasticity on local VIP neurons may provide an additional motif for regulating cortical network function in addition to pyramidal cell excitatory and inhibitory synaptic plasticity.

We used the VIP-Cre transgenic mouse line[48] to express GFP in a sparse subset of superficially located VIP neurons in the somatosensory cortex. To obtain sparse expression we utilized Cre-dependent AAV-vectors encoding GFP (Fig. 1). Immunofluorescence indicates that this approach drives GFP expression in VIP neurons in a specific manner (Supplementary Fig. 1, 3–6). Despite limitations of immunolabeling for estimating gene expression, the fraction of GFP-expressing cells that were clearly positive for VIP was similar to previous studies that utilized Cre-dependent AAV expression vectors[37,48]. None of the GFP-expressing neurons were co-labeled with anti-PV or anti-SST antibodies, which makes it unlikely that expression had leaked into one of the other main inhibitory cell types. In addition, expression of GFP was most likely not restricted to one particular subtype of VIP neurons, since various fractions of cells co-expressed other, well-characterized, VIP cell markers, such as CR, NPY or CCK.

Which cell type gives rise to the spiny dendrites that we observed in L1? 3D reconstructions of the imaged spiny neurons indicated that their somata were predominantly located at the interface between L1 and L2; and that they had a multipolar dendritic morphology (Fig. 1), resembling spiny VIP-positive neurons that have previously been described[44] or L2/3 non-fast spiking double bouquet or small basket cells[22]. These cells may be similar to a subset of spiny multipolar interneurons in the hippocampal CA1 area of GAD65-GFP mice that were found to express VIP[24]. Spiny GFP-positive cells that were retrospectively re-identified in brain sections, or GFP-positive cells in brain sections with spiny peri-somatic dendrites were incidentally co-labeled with anti-CCK or anti-CR antibodies (Supplementary Fig. 5 and 6), which indicates that they may span various VIP cell subtypes close to L1[52,55,57].

The electrophysiological measurements corroborate the morphological characteristics (Fig. 2). We found that the spiny neurons tended to fire action potentials in bursts, and that their intrinsic electrophysiological properties are similar to small VIP subtypes that have been identified in other studies, which were also located close to L1[49,50], and often express CCK[50]. In recent transcriptomic studies, such cells have been classified as synuclein-expressing (SNCG) VIP neurons[51,72,73].

We found that different VIP cell dendritic segments had highly variable spine densities (e.g. varying from 0.4 to 1.3 μm⁻¹; Fig. 3). In very spiny parts of the dendrites (~0.8 μm⁻¹, Fig. 1), mean

**Table 2 Synapse data.**

| Type | Dend L (μm) | # Syn Dens (μm$^{-1}$) | # Exc Syn Dens (μm$^{-1}$) | SA Exc Syn ± s.d. (μm$^2$)* | # Inh Syn Dens (μm$^{-1}$) | SA Inh syn ± s.d. (μm$^2$) | # Uncl Syn Dens (μm$^{-1}$) | # Exc Syn on spines | # Inh Syn on spines | # Comp Syn |
|---|---|---|---|---|---|---|---|---|---|---|
| VIP | 161 | 309 1.92 | 248 [80%] 1.54 | 0.13 ± 0.08 | 30 [10%] 0.19 | 0.12 ± 0.07 | 31 [10%] 0.19 | 68 [27%] | 7 [23%] | 58 [19%] |
| CTR | 67 | 123 1.84 | 97 [79%] 1.45 | 0.19 ± 0.15 | 23 [19%] 0.34 | 0.13 ± 0.07 | 3 [2%] 0.04 | 73 [75%] | 3 [13%] | 6 [5%] |

*Dend L* dendritic length, *#* number, *Syn* synapse, *Dens* density, *Exc* excitatory, *Inh* inhibitory, *Uncl* unclassified, *SA* surface area, *Comp* compound, *s.d.* standard deviation, * significant difference (see Fig. 6).

densities were typically higher than what has been described for other cortical inhibitory neurons[22,28]. However, it should be noted that we restricted our imaging to distal parts of the apical dendrites that appeared more spiny than proximal dendrites. Uneven spine distributions have been described for cortical SST-expressing interneurons[74] and hippocampal NPY or CR-expressing interneurons[24,75]. In this respect, interneurons are quite distinct from pyramidal neurons, which usually display equal spine densities along their dendrites[76].

Over various imaging sessions at 4-day intervals we observed stable, disappearing and appearing spines (Fig. 3), similar to what has previously been described for pyramidal neurons[9] and other types of cortical interneurons in vivo[28]. However, spine dynamics seemed to be higher than what had been observed before on excitatory or inhibitory neurons in the mouse neocortex. We found that over any 4-day interval approximately 30% of the spine population of VIP spines had disappeared or newly appeared, which is higher than the average 15-20% TORs over 1 to 7 days that were typically observed for pyramidal cell spines[60,77]. Similarly, the decline in the VIP spine survival fractions over time was steeper than what is generally observed for pyramidal spines. We found that the VIP spine population was proportionally much less stable than those of L2/3 pyramidal neurons (Fig. 4).

Altogether, our data indicate that VIP neurons with spiny dendrites in L1 display higher spine turnover rates than cortical pyramidal neurons, and other inhibitory neurons described so far[9,10,12], and approach the high spine dynamics that have recently been estimated for pyramidal neurons in the hippocampus, which are hypothesized to subsist the volatile memory functions of this area[78,79]. This suggests that VIP neurons change their connectivity at a higher rate than cortical pyramidal neurons and may over time receive input from a vastly larger number of axonal boutons in L1. This might be functionally similar to the high dendritic branch remodeling on other types of interneurons[26,80,81]. The rapid spine alterations may endow interneurons with a rapid response capacity to changes in sensory input, and drive homeostatic and experience-dependent plasticity of cortical networks[23,28,82].

Using SBEM we found that VIP spines are relatively small as compared to spines on a nearby presumptive pyramidal dendrite (Fig. 6). The small sizes may be the result of the relatively short average life times, since new spines grow in size relatively slowly[2,83] and because further spine volume changes are proportional to their size[83,84]. Yet most reconstructed new spines had synapses (Figs. 5 and 6) and although they were unstable, 50% tended to remain present for at least 4 days (Fig. 3). New spines on pyramidal cells typically live shorter and do not readily form synapses[2,12]. This suggests that whereas spines on pyramidal neurons are thought to selectively 'sample' the environment for suitable synaptic input but largely fail at forming synapses[9,10], new VIP spines may more readily connect to any axon or bouton that they encounter. This may cause them to have a relatively longer 'transient' life span.

Ultrastructural analysis revealed various other striking features of spiny VIP dendrites. A substantial portion of their spines had more than 1 asymmetrical excitatory synapse. The high number of inputs per spine might be a particular feature of VIP spines, since reported synapse per spine ratios in other inhibitory neurons in the cortex or hippocampus are typically lower[22,24,74], but see[75]. Furthermore, the majority of VIP (~79%) spines were found to contain ER, similar to CTR spines (~76%). ER occurs mostly in mature spines (~70% of mature spines and ~30% of new spines with synapses)[2], and can function as an internal calcium store[85]. Therefore, VIP spines have at least a similarly high capacity to store and release calcium as compared to pyramidal cell spines. It is also interesting to note that at least half of the new spines on the reconstructed VIP dendrite contained ER. This confirms the earlier notion that new spines on VIP dendrites may rapidly acquire a mature phenotype after they have been formed, which is distinct from the rather slow or complete lack of maturation of most new spines on pyramidal neurons.

We also found that some of the shaft synapses, as well as some of the synapses on stable and new spines were with multisynaptic boutons (MSB) or boutons of the same axon, which is in line with observations in pyramidal neurons[2,70,71]. This also supports the possibility that spines make synaptic connections with pre-existing boutons rather than inducing *de novo* bouton formation, as has been hypothesized previously[2,86].

The large fraction of compound or redundant synaptic connections on VIP dendrites (18%) is comparable to fractions found on CA1 pyramidal cell branches in the stratum lacunosum-moleculare[70]. In that study, computational simulations indicated that such compound connections efficiently depolarize distal dendrites. In addition, we found that VIP dendrites have a high prevalence for spines with multiple excitatory synapses (16%) and have a relatively low fraction of inhibitory inputs (10–20%). The latter fraction is lower than what has been found on hippocampal VIP-like interneurons, although inhibitory synapse density typically run down towards the distal ends of dendrites[87,88]. These three factors together suggest that spiny VIP dendrites are efficiently depolarized by clustered synaptic inputs, and may readily spike upon activation of a relatively low number of afferents.

What could be the role of this subtype of VIP neurons in the cortical circuit? The location of their dendrites in L1 suggests that they receive long-range excitatory synaptic input[89], yet BS multipolar cells have also been shown to receive relatively strong excitatory inputs from L2/3 and L5[90]. A large proportion of the BS VIP cell population is thought to mediate disinhibition[32,40,41,45,91–94]. Although the exact role of the small BS type of VIP neurons has yet to be described, immunofluorescence studies suggest that they may exert disinhibition by preferentially targeting PV interneurons[95]. Since we found that VIP spines have low survival rates yet rapidly form synapses once they are generated, this may imply that these neurons do not serve as stable intermediates for cortical disinhibition, but are able to rapidly switch between various and perhaps distinct synaptic inputs. They may thereby serve as mediators

of rapid changes in the excitatory and inhibitory balance in cortical circuits. Such dynamics match well with their suggested role in cortical disinhibition for diverse cognitive functions such as motor control and sensory discrimination[37,38,45].

## Methods

**Mice and viral vectors**. All experiments were performed in accordance with the guidelines of the Swiss Federal Act on Animal Protection and Swiss Animal Protection Ordinance. The ethics committee of the University of Geneva and the Cantonal Veterinary Office (Geneva, Switzerland) approved all experiments (licenses GE/28/14, GE/61/17, and GE/121/19).

Experiments were performed in VIP-IRES-Cre [(VIPtm1(cre)Zjh] mice ($N = 5$) that were obtained from The Jackson Laboratory. In this knock-in mouse line (Viptm1(cre)Zjh/J) that was originally generated by the Huang laboratory in Cold Spring Harbor[48], Cre recombinase is directed to VIP-expressing neurons by the endogenous promoter/enhancer elements of the VIP genomic locus. The knock-in allele contains an internal ribosome entry site (IRES) that allows for independent translation initiation of Cre recombinase. Previous studies have reported Cre-recombinase activity specifically in VIP-expressing neurons[37,48,96].

To express GFP for in vivo imaging experiments, Cre-dependent AAV viral vectors were used (AAV9.CAG.flex.eGFP.WPRE.bGH [AllenInsitute854] acquired from the Penn Vector core; based on plasmid, AAV pCAG-FLEX-EGFP-WPRE, by Hongkui Zeng (Addgene plasmid # 51502)[97]. In these vectors the expression of eGFP is directed by a CMV-enhanced beta-actin (CAG) promoter upon Cre-mediated inversion of the GFP open reading frame. To label cells for electrophysiology, we used AAV2.hSyn.DIO.mCherry (by Brian Roth, Addgene plasmid #50459); and for immunofluorescence in Supplementary Figure 6, we used AAV2.hSyn.DIO.EGFP (by Brian Roth, Addgene plasmid #50457).

**Cranial window implantation**. At the time of surgery, the animals were at least 4 weeks old. Cranial window implantation was performed following the procedure described previously[77]. In brief, animals were anesthetized by an intraperitoneal (IP) injection of a mixture of anesthetics and sedatives (MMF), consisting of Medetomidin (0.21 mg kg$^{-1}$ body weight), Midazolam (5 mg kg$^{-1}$), and Fentanyl (0.05 mg kg$^{-1}$). A craniotomy was created by removal of a circular piece of the bone above the barrel cortex (3.5 mm lateral and 1.5 mm caudal from bregma) on the right hemisphere. In order to achieve very sparse expression in a small subset of neurons a minimal amount of virus (in the order of 5–10 nL) was injected using a glass pipette (Wiretrol, Drummond), attached to an oil hydraulic manipulator (MMO-220A, Narishige). The pipette was loaded with a drop of virus and rapidly punched into the brain to a depth of 100–150 μm below the dura. After the injection, the craniotomy was covered by a 3 mm diameter glass window, which was glued to the bone using acrylic dental cement (JET REPAIR, Lang Dental Mfg) and acrylate glue (ULTRA GEL, Pattex). Apart from the penetration by the injection pipette, the dura was left intact. After the surgery, the mice were awoken by an IP injection of a mixture of anti-sedatives and analgesics (atipamezole (2.5 mg kg$^{-1}$), flumazenil (0.5 kg$^{-1}$), naloxone (0.1 kg$^{-1}$)). Animals also received a mixture of buprenorphine (0.1 μg g$^{-1}$) and Carprofen (5 μg g$^{-1}$) to reduce pain and inflammation. Imaging started 2-3 weeks after the cranial window implantation.

**In vivo imaging**. In vivo images were acquired under MMF-anesthesia, using a custom-built 2PLSM (Janelia Farm Research Campus, model Non-MIMMS in vivo microscope[77]. To record the 2PLSM images, we used custom software written in MATLAB (Scanimage, Vidriotech and Janelia Farm Research Campus). A pulsed Ti:sapphire laser (Chameleon ultra II, Coherent) was tuned to 910 nm to excite the GFP. The green channel was equipped with a photomultiplier tube (Hamamatsu H10770PA-40SEL). We used a 20x water immersion objective (NA 0.95, XLUMPlanFI, Olympus, Japan), and images were acquired at 2 ms/line (image size 1024 × 1024 pixels, pixel size 0.87 μm², z-plane steps: 2 μm for large z-stacks used for exploration of the preparation and the Neurolucida reconstructions, and 1024 × 1024 pixels z-stacks with a pixel size of 0.0049 μm² and z-steps of 1 μm, for the acquisition of images that were used for spine tracking). Neurolucida (MBF Bioscience) software was used to trace the 3D morphology in the 2PLSM image stacks (pixel size 0.87 μm²) 2PLSM z-stacks.

**Immunofluorescence**. For immunofluorescence, the animals were anaesthetized and fixed using transcardial perfusion of 4% paraformaldehyde (PFA) in saline. The brains were collected and left in 4% PFA overnight (4 °C) for further fixation. Tangential or coronal brain sections (50 μm thickness) were obtained using a vibratome (Leica VT1200S; Leica Microsystems, Vienna, Austria) and initially stored in PBS. The sections were incubated in a blocking solution (1% BSA, 10% Donkey serum, 1% Triton X-100 in PBS) for 2 hours at room temperature. Sections were then rinsed (3x 10 min) and incubated in a diluted (1:10) blocking solution containing the primary antibodies overnight at 4 °C. The sections were thoroughly rinsed (6x 10 min) and then incubated with secondary antibodies for 2 h at room temperature. The following antibodies were used: goat anti-parvalbumin (1:2000, Swant, PVG-214), rabbit anti-vasoactive intestinal polypeptide (1:400,

Immunostar, #20077), rat anti-somatostatin (1:100, MerckMillipore, MAB354), goat anti-calretinin (1:5000, Swant, CG1), rabbit anti-neuropeptide Y (1:500, Abcam, ab10980), and rabbit anti-cholecystokinin (1:500, Sigma, SAB2100357). As secondary antibodies, we used alexa 568/647 anti-rabbit, alexa 568 anti-rat, and alexa 568 anti-goat (Thermofisher, 1:200). Sections were mounted on glass using VECTASHIELD mounting medium (H-1400).

Images were taken using a Zeiss LSM 710 META confocal microscope. Lasers were tuned to 488 nm and 555 nm at 10 mW and at 639 nm at 5 mW and signals were filtered using conventional mounted filters that allowed minimal bleed-through between the channels, BP575-615IR for the A568, BP505-530 for the GFP, and LP640 for A647. The objectives were a Plan-Neofluar 20x (0.50) and 40x (1.3 oil immersion).

Image analysis on these images was performed in Fiji[98]. Regions of interest (ROI) around the cell bodies were algorithmically created by thresholding the fluorescence of GFP-expressing cells in the green channel, using the Kapur-Sahoo-Wong (maximum entropy) method[99]. The resulting masks were blurred with a 2-d Gaussian ($\sigma = 2.5$ μm) and binarized with the same threshold. In rare cases of overlap between neighboring cells, the ROIs were segmented using the watershed algorithm. ROIs smaller than 25 μm² were discarded. The integral of brightness above threshold inside a cell's ROI, the percentage of pixels inside the ROI above threshold, and the minimum brightness above threshold of the ROI for all imaging channels were calculated.

**Electrophysiology**. Mice were anesthetized with isoflurane (1–4% mixed in oxygen) and a subsequent intraperitoneal injection of a ketamine (100 mg/kg)/xylazine (10 mg/kg) cocktail. They were perfused through the heart with ice-cold saline consisting of (in mM): 2.5 KCl, 1.25 NaH2PO4, 25 NaHCO3, 0.5 CaCl2, 7 MgCl2, 7 dextrose, 205 sucrose, 1.3 ascorbate and 3 sodium pyruvate (bubbled with 95% O2/5% CO2 to maintain pH at ~7.4). A vibrating tissue slicer (Leica VT S1000, Germany) was used to prepare 300 μm thick coronal sections of the primary somatosensory cortex. Acute slices were held for 30 minutes at 37 °C in a chamber filled with artificial cerebral spinal fluid (aCSF) consisting of (in mM): 125 NaCl, 2.5 KCl, 1.25 NaH2PO4, 25 NaHCO3, 2 CaCl2, 2 MgCl2, 10 dextrose and 3 sodium pyruvate (bubbled with 95% O2/5% CO2) and then at room temperature for at least 30 minutes.

Electrophysiological recordings were made from multipolar VIP positive neurons located in the superficial layers of the primary somatosensory cortex (barrel cortex). Slices were placed in a heated (32–34 °C) recording chamber that was continually perfused (1–2 138 mL/minute) with aCSF containing (in mM): 124 NaCl, 3 KCl, 2 CaCl2, 1.3 MgSO4, 26 NaHCO3, 1.25 NaH2PO4, 10 D-glucose with an osmolarity of 300 mOsm and pH: 7.4, bubbled with 95% O2-5% CO2. Fluorescent neurons were identified using a coolLED pE-300ultra illumination system and a 540/605 nm excitation/emission filter set. Patch pipettes (5-7 MΩ) were pulled from borosilicate glass. The pipette solution for all configurations contained (in mM): 110 K-gluconate, 10 KCl, 10 HEPES, 4 ATP, 0.3 GTP, 10 phosphocreatine and 0.4% biocytin and pH between 7.2 and 7.3. Biocytin (Sigma; 0.4%) was also included for histological processing.

Data were acquired using a 200B amplifier that was interfaced to pClamp command/record software through a Digidata 1440 A analog/digital converter (Axon Instruments; low-pass filter = 10 kHz, sampling rate = 100 kHz). Pipette capacitance was compensated, and the bridge was balanced during each recording. Series resistance was monitored throughout each experiment and was 8.9 ± 0.4 MΩ. Accepted deviations from these parameters in current transients recorded over the time-windows used for statistical analysis were less than 20%. Voltages were not corrected for the liquid-junction potential (estimated as ~11 mV). Data were analyzed using Clampfit 10.6 and GraphPad Prism. Input resistance (RN) and series resistance was monitored by measuring passive current transients induced by 10 mV hyperpolarizing voltage steps from a holding potential of −70 mV. The transients were reliably fitted with a bi-exponential function (tau fast and tau slow). The sag was calculated as the difference between the maximum negative voltage response and the steady state during hyperpolarization current injection of −100 pA. The rebound depolarization amplitude was calculated as the difference between the steady state potential and the maximum depolarization potential. After-hyperpolarization potential (AHP) was calculated as the hyperpolarizing potential that falls below the action potential (AP) rheobase potential in its repolarizing phase. Depolarizing after potentials (ADP) was calculated as the peak depolarizing potential following the AHP. Interspike interval (ISI) was calculated as the time difference between the peaks of two consecutive APs. For detecting the presence of burst, the ISI between the first two APs was calculated at rheobase. For the accommodation index the ratio between the 3 ISIs following the one analyzed for the bursting detection and the last 3 ISIs at rheobase was calculated. Mann Whitney tests were used to statistically compare the two groups.

After the recordings, slices were fixed in 4% paraformaldehyde in 0.1 M PBS for 20 minutes. The reaction was performed using the ABC-Elite kit (Vector Laboratories) and the DAB kit (Vector Laboratories). After the reaction, slices were mounted on a coverslip and used for the morphological reconstruction in Neurolucida (MBF Bioscience) using an upright microscope (Olympus BX51) equipped with a 100X immersion-oil objective.

**Electron microscopy**. Immediately after the final imaging session, animals were perfused, via the heart, with a buffered mix of 2.5% glutaraldehyde and 2.0 % paraformaldehyde in 0.1 M phosphate buffer (pH 7.4), and then left for a further 2 hours. The brain was then removed and vibratome sections were cut at 60–80 microns thickness tangential to the surface of the imaged region of the cortex.

For FIBSEM: The sections were washed thoroughly with cacodylate buffer (0.1 M, pH 7.4), postfixed for 40 minutes in 1.0 % osmium tetroxide with 1.5% potassium ferrocyanide, followed by 40 minutes in 1.0% osmium tetroxide alone. They were finally stained for 30 minutes in 1% uranyl acetate in water before being dehydrated through increasing concentrations of alcohol and then embedded in Durcupan ACM (Fluka, Switzerland) resin and left to harden for 24 hours in a 65 °C oven.

For SBM: These sections were stained and embedded according to a protocol similar to Hua et al (2015)[100]. Briefly, the sections were post-fixed in potassium ferrocyanide (1.5%) and osmium (2%), then stained with thiocarbohydrazide (1%) followed by osmium tetroxide (2%). They were then stained overnight in uranyl acetate (1%), washed in distilled water at 50 °C, before being stained with lead aspartate at the same temperature. They were finally dehydrated in increasing concentrations of alcohol and then embedded in Spurr's resin and hardened at 65 °C for 24 h between glass slides.

Low-magnification brightfield images of the sections were aligned using their vasculature pattern and overlaid with the in vivo images of cortical surface, taken through the cranial window. This allowed us to identify the region of interest. Laser fiducial marking[62,101] was performed by tuning the laser to 810 nm at its maximal power (approximately 300 mW at the back focal plane of the objective). 512 to 2500 line-scans (2 ms/line) were used to brand a line-mark close to the dendrite. A square area of approximately $25 \times 25$ µm$^2$ was demarcated by such laser burn marks. After laser branding the section was processed for EM as described previously[102].

The region of the section containing the neurites of interest was trimmed from the rest of the section and glued to an aluminium stub using conductive resin. This was then trimmed with a glass knife and the final block mounted either inside a block face scanning electron microscope (SBEM) consisting of a scanning electron microscope (Zeiss Merlin, Zeiss NTS) containing an ultramicrotome (3View, Gatan) or a focused ion beam scanning electron microscope (FIBSEM) (NVision 40, Zeiss NTS). For FIBSEM, the block face was imaged at 1.7 kV with a current of 2.2 nA, a pixel size of 6 nm and slice thickness of 12 nm. The field of view was $18 \times 29$ µm. A total stack of 1580 images was acquired. In SBEM, layers of resin, 50 nm thick, were cut from the block surface, and sequential images collected after each layer had been removed. An acceleration voltage of the 1.7 kV was used with a pixel size of 6.5 nm with a dwell time of 1 µs. A total of 600 slices of 50 nm thickness were sectioned from the block and tiled images taken after each cut. The total field of view imaged was $105 \times 109$ µm. Images were aligned in the FIJI imaging software (www.fiji.sc)[98], and the segmentation of the dendrites and axons was carried out by hand using the TrakEM2 software operating in FIJI[103]. The final models were exported to the 3D modelling software (Blender.org) for analysis using the Neuromorph toolset[104].

We reconstructed 3 VIP dendritic segments (1 in FIBSEM, 2 in SBEM) and 1 control (CTR) dendritic segment (SBEM). In the results and figures, numeric data of spines and synapses are expressed as the mean ± standard deviations (sd). Statistical comparisons between the VIP and CTR dendritic segments were performed using unpaired two-tailed Student's t-tests.

**Spine analysis**. Spine annotation and correlation between sessions was performed using custom software written in MATLAB (part of Scanimage v3.5, Janelia Farm Research Campus). Linear spine density was calculated as the number of spines per dendritic unit length. The spine turnover ratio (TOR) was calculated as:

$$R_{(t_i, t_j)} = \left( N^{gained}_{(t_i, t_j)} + N^{lost}_{(t_i, t_j)} \right) / \left( N_{t_i} + N_{t_j} \right) \qquad (3)$$

where $t_i, t_j$ are two consecutive time points, $N_{t_i}$ the total number of spines at $t_i$, $N^{gained}_{(t_i, t_j)}$ and $N^{lost}_{(t_i, t_j)}$ the number of gained or lost spines between $t_i$ and $t_j$. The spine survival fraction (SF) was calculated as:

$$SF_{t_i} = \left( N_0 - N^{lost}_{(0, t_i)} \right) / N_0 \qquad (4)$$

where $t_i$ is time point $i$, $N_0$ the total number of spines on the first day of imaging ($t_0$), and $N^{lost}_{(0, t_i)}$ the number of spines that was lost until $t_i$ relative to $t_0$.

**Statistical models for immunofluorescence and spine dynamics**. A generative model was built to estimate the percentage of positive cells for different antibody stainings obtained with immunofluorescence analysis.

$$o_i \sim Bernoulli(p_i)$$
$$logit(p_i) = \alpha_{slice} + b_d X_{id} + \sum_j \beta_j \, Y_{ij} + \sum_k ln(Z_{ik})$$
$$\alpha_{slice} \sim Normal(0, \sigma_{slice})$$
$$\sigma_{slice} \sim StudentT(3, 0, 10)$$
$$\beta_j \sim Normal(0, 1)$$
$$b_{VIP} \sim Normal(3, 1)$$
$$b_{PV} \sim Normal(-3, 1) \qquad (5)$$
$$b_{NPY} \sim Normal(-2.4, 1)$$
$$b_{CCK} \sim Normal(-1.61, 1)$$
$$b_{CR} \sim Normal(0, 1)$$
$$b_{SST} \sim Normal(-2.1, 1)$$

The binary outcome variable $o_i$, corresponding to whether a cell was manually scored as positive for a given staining or not, was modeled as drawn from the Bernoulli distribution. The probability of the binary outcome was mapped to a linear scale through the logit link function,

$$logit(p) = \log \frac{p}{1-p} \qquad (6)$$

The variability of the fluorescence signal at the level of individual slices was modeled as a varying intercept, $\alpha_{slice}$. A parameter $b_d$, was associated with the truth value of each molecular marker, $X_d$, where $d \in \{VIP, PV, NPY, CCK, CR, SST\}$, which evaluated to true when the slice was labeled with the corresponding antibody. A prior probability was assigned to the coefficient of each molecular marker according to the percentages of overlap with VIP cells reported in the literature (when transformed to the probability scale): VIP: 0.95, PV: 0.05, NPY: 0.08, CCK: 0.17, CR: 0.5, SST: 0.11[36,54,56]. In cases where a complete absence or presence of overlap was expected, as was the case for PV and VIP, we assigned a 0.05 or 0.95 probability, respectively, to avoid infinite values on the linear scale. For the prior probability of overlap for SST with VIP, an expected probability of 0.11 was assigned, which corresponds to the geometric mean of the values reported in the literature, after replacing 0 probability with 0.05 [0.5, 0.05, 0.05][36,54,56]. A set of variables $Y_{ij}$, and their corresponding coefficients, $\beta_j$, was included in the model as predictors for each antibody labeling channel: (i) the integral of brightness above threshold inside a cell's ROI, (ii) the percentage of pixels inside the ROI above threshold, (iii) the minimum brightness above threshold of the ROI, (iv) the mean brightness of the ROI, (v) all possible two- and three-way interactions of the previous predictors. The ROI's mean brightness, the percentage of the ROI's area above threshold, and the ROI's integral of brightness measured from the GFP channel of each ROI were included as offsets, $\sum_k ln (Z_{ik})$, to further control for variation unrelated to antibody labeling. The posterior distribution of the model parameters was estimated using R[105], and package brms[106] as an interface to the probabilistic modeling language Stan[107].

The spine densities and TOR were fitted with linear mixed-effects models with time as the fixed effect, and dendritic ID and mouse ID as random effects, using the restricted maximum likelihood estimation method (MATLAB function fitlme). A type III F-test on the residuals of the fit was performed in order to assess marginal differences, using the Satterthwaite approximation for estimating the degrees of freedom (MATLAB function anova). F-values and degrees of freedom are reported.

To facilitate comparison with previous studies on spine survival, we fitted, using the non-linear least squares method (R function nls), the spine survival fractions with single exponential ($ae^{bt}$) and two-exponential sum functions ($ae^{bt} + ce^{dt}$), where $a, b, c, d$ are free parameters and $t$ is time. We performed model comparison using the corrected Akaike Information Criterion (cAIC) which is given by:

$$AIC = -2log(\hat{L}) + 2k \quad \text{(Akaike Information Criterion)} \qquad (7)$$

$$cAIC = AIC + \frac{2k(k+1)}{n-k-1} \quad \text{(corrected AIC)} \qquad (8)$$

where $\hat{L}$ the estimated likelihood of a model, $k$ the number of estimated parameters of the model, $n$ the number of observations. The Akaike Information Criterion[108] offers an estimate of the information lost when fitting a given model while taking into account the number of parameters. The model with the lowest AIC offers the best compromise between goodness-of-fit and complexity of the model. The cAIC includes a correction for finite samples[109]. A log-ratio test also gave equivalent results to those obtained with the cAIC in all cases.

To compare survival rates between spines of VIP and pyramidal neurons (L 2/3 neurons from a previous study[60]), we fitted the data with a Cox proportional hazards model[110], a semi-parametric method of estimating hazard rates adjusted for cell type:

$$h(x_i, t) = h_0(t)e^{bx_i} \qquad (9)$$

where $X_i$ is the predictor of the $i$th spine, $x_i \in [VIP, PYR]$. The term $h_0(t)$ represents the baseline hazard rate relative to 0 for mouse c, while $b$ is the coefficient estimate of the hazard ratio. Only the baseline term $h_0(t)$ depends on

time and the coefficient *b* expresses the log-linear relationship of the hazard ratio for the two levels of the predictor variable. Robust standard errors (Huber-White) adjusting for the correlations between spines from the same mouse are reported. Hypothesis testing against the null of a common hazard rate with the Wald test or the likelihood ratio test provided equivalent results.

For all statistics, the tests are indicated in the results or methods, and unless otherwise stated, the standard error of the mean (electrophysiology, immunofluorescence, parameters estimated from statistical models) or standard deviation (spine dynamics, electron microscopy) was used as an estimate of dispersion.

**Reporting summary**. Further information on research design is available in the Nature Research Reporting Summary linked to this article.

## Data availability
Data used to generate the figures will be available upon request or will be made freely available at the CERN data repository Zenodo https://zenodo.org/communities/holtmaat-lab-data/ upon publication of the manuscript.

## Code availability
Code will be available upon request or will be made freely available at the CERN data repository Zenodo https://zenodo.org/communities/holtmaat-lab-data/ upon publication of the manuscript.

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

## Acknowledgements

We thank Marco Cantoni and Anaelle Dubois for help with EM imaging, Stephanie Clerc-Rosset for the preparation of samples for EM, Vanessa Schubert for providing data on L2/3 pyramidal cells, Foivos Morkopoulos and Roberta Leone for comments and discussion on the epyhs data, and Ronan Chereau and Leena Williams for comments on the

manuscript. This work was supported by the Swiss National Science Foundation (grants # 31003A_153448 and 31003A_173125 to A.H., and #CRSII3_154453 to A.H. and G.K.), the Marie Skłodowska-Curie Actions (grant #895465 to V.K.), the Initial Training Networks Projects Marie Curie Actions, NPLAST (A.H., C.G.), and the International Foundation for Research in paraplegia (Chair Alain Rossier to A.H.).

## Author contributions

C.G., V.K., K.S.L., F.B., G.K. and A.H. designed the experiments. C.G., V.K., K.S.L. and A.H. collected and analyzed in vivo imaging data. C.G., V.K. and K.S.L. collected and analyzed immunofluorescence data. C.G. made neurolucida reconstructions. F.B. collected and analyzed the electrophysiology data. V.K. built statistical models for immunofluorescence and spine dynamics. D.A.S., J.B. and G.K. collected EM data. D.A.S., J.B., G.K. and A.H. made the EM reconstructions and analyzed the EM data. A.H. provided equipment and technical expertise for the in vivo imaging. G.K. provided equipment and technical expertise for the EM. G.K. and A.H. supervised the research. All authors provided figure material. A.H. produced figures and wrote the manuscript. All authors edited the manuscript.

## Competing interests

The authors declare no competing interests.
