## [Peer Review File · Communications Biology]

This manuscript has been previously reviewed at another Nature Portfolio journal. This document only contains reviewer comments and rebuttal letters for versions considered at Communications Biology.

Reviewers' comments:

Reviewer #1 (Remarks to the Author):

In the revised MS, the Authors attempted to clarify the types of VIP interneurons with spiny dendrites. Based on the results obtained by immunostaining and in vitro electrophysiology, the Authors suggest that spiny VIP interneurons are small basket cells. This suggestion is particularly interesting as the Authors themselves wrote in the text they do not have any proof for that.

I would draw the Authors' attention to the fact that the existence of small basket cells in L1 has not been supported by any evidence; the small basket cell category is only an assumption at present. If the Authors carefully read the papers cited (Posluszny, 2019, Tremblay et al., 2016 etc.), then they may notice that the small basket cell in L1/L2 is a presumed category only, indicated often with a question mark, without any evidence that these interneurons are indeed basket cells targeting predominantly the perisomatic region of pyramidal neurons. If the Authors favor the idea that spiny VIP interneurons are small basket cells, then they should confirm this hypothesis by investigating the target distribution of biocytin-filled interneurons sampled in slices. Alternatively, they may test the CB1 cannabinoid receptor expression on the axon terminals of biocytin-filled spiny VIP interneurons using immunostaining. As it is known that the perisomatic innervation of pyramidal cells originates primarily from basket cells expressing either parvalbumin or CB1 receptors and small basket cells are proposed to lack parvalbumin, small basket cells should express cannabinoid receptors on their axonal boutons. Either approach could prove that spiny VIP interneurons are basket cells. Otherwise, using small basket cell category for description of an interneuron type without any proof is misleading.

But taking into account that only 6% of genetically labeled VIP interneurons contained CCK (see Results), a cell group that may be composed of basket cells and other interneuron types, as CCK is not a definite marker for basket cells (Rovira-Esteban et al., 2019), the probability of investigating the dynamics of spines on basket cell dendrites expressing VIP is very low. The chance, however, is much higher that the Authors have studied the spiny dendrites of VIP interneurons expressing calretinin (~40% of VIP interneurons expressed CR, see Results), typically present in interneuron selective interneurons. The spiking features of spiny VIP neurons shown in Fig. 2 is actually resemble those described earlier for VIP/CR interneurons (He et al., 2016).

Though, the Authors show that spines are present on the proximal dendrites of some interneurons, it would be more accurate to show the spines on the distal dendrites of the biocytin-filled and neurochemically identified interneurons, as the spine dynamics were evaluated in L1, where the distal dendrites of VIP neurons arborize.

Minor:

The properties of EPSCs have been evaluated in spiny and non-spiny neurons (Fig. 2). However, it is not clear how EPSCs were separated from IPSCs. The recordings were obtained at a holding potential of -70 mV, and the intrapipette solution contained 24 mM Cl⁻, leading to the reversal potential for Cl⁻ around -50 mV. This configuration predicts that spontaneous PSCs recorded at -70 mV should be a mixture of EPSCs and IPSCs, as a portion of inward currents may originate from opening of GABA-A receptors.

The Authors claim in the text that 80% of synapses on VIP dendrites were excitatory, a ratio that fits to the general ratio of excitatory and inhibitory neurons in cortical structures. However, earlier EM studies showed that inputs on interneurons expressing CR (and likely VIP) and CCK, receive a disproportionately higher number of inhibitory synapses as the general ratio of excitatory and inhibitory cells predicts (Gulyas et al., 1999; Matyas et al., 2004). This discrepancy may deserve discussion.

Reviewer #2 (Remarks to the Author):

I am satisfied with the changes the authors have made to the manuscript. This is a great study!

Reviewers' comments:

Reviewer #1 (Remarks to the Author):

In the revised MS, the Authors attempted to clarify the types of VIP interneurons with spiny dendrites. Based on the results obtained by immunostaining and in vitro electrophysiology, the Authors suggest that spiny VIP interneurons are small basket cells. This suggestion is particularly interesting as the Authors themselves wrote in the text they do not have any proof for that.

I would draw the Authors' attention to the fact that the existence of small basket cells in L1 has not been supported by any evidence; the small basket cell category is only an assumption at present. If the Authors carefully read the papers cited (Posluszny, 2019, Tremblay et al., 2016 etc.), then they may notice that the small basket cell in L1/L2 is a presumed category only, indicated often with a question mark, without any evidence that these interneurons are indeed basket cells targeting predominantly the perisomatic region of pyramidal neurons. If the Authors favor the idea that spiny VIP interneurons are small basket cells, then they should confirm this hypothesis by investigating the target distribution of biocytin-filled interneurons sampled in slices. Alternatively, they may test the CB1 cannabinoid receptor expression on the axon terminals of biocytin-filled spiny VIP interneurons using immunostaining. As it is known that the perisomatic innervation of pyramidal cells originates primarily from basket cells expressing either parvalbumin or CB1 receptors and small basket cells are proposed to lack parvalbumin, small basket cells should express cannabinoid receptors on their axonal boutons. Either approach could prove that spiny VIP interneurons are basket cells. Otherwise, using small basket cell category for description of an interneuron type without any proof is misleading.

But taking into account that only 6% of genetically labeled VIP interneurons contained CCK (see Results), a cell group that may be composed of basket cells and other interneuron types, as CCK is not a definite marker for basket cells (Rovira-Esteban et al., 2019), the probability of investigating the dynamics of spines on basket cell dendrites expressing VIP is very low. The chance, however, is much higher than the Authors have studied the spiny dendrites of VIP interneurons expressing calretinin (~40% of VIP interneurons expressed CR, see Results), typically present in interneuron selective interneurons. The spiking features of spiny VIP neurons shown in Fig. 2 actually resemble those described earlier for VIP/CR interneurons (He et al., 2016).

Though, the Authors show that spines are present on the proximal dendrites of some interneurons, it would be more accurate to show the spines on the distal dendrites of the biocytin-filled and neurochemically identified interneurons, as the spine dynamics were evaluated in L1, where the distal dendrites of VIP neurons arborize.

We thank the reviewer for these comments. We agree with many of these points. In essence, our conclusion that spiny VIP cells are predominantly CCK-expressing small basket cells was perhaps a bit premature and not fully (quantitatively) supported by our own experimental data or literature.

To get a clearer idea as to whether the spiny VIP cells could represent a particular subtype with a common cytochemical makeup, we decided to try and directly assess the expression of two marker proteins (CR and CCK) in spiny cells that were first identified in vivo. Sparsely GFP-expressing VIP dendrites were imaged through a cranial window in two mice. Perfusion-fixed brain sections were subsequently immunostained for the two markers. We managed to re-identify 11 in vivo-imaged neurons in the immunofluorescence confocal images and found 4 neurons with high spine densities. One of those was positive for CCK, one for CR and two for none of these markers. This data set has been added as Supplementary Figure 6 to the manuscript. Even though this experiment may not

provide statistically testable data, it does show that the population of spiny VIP neurons may include a variety of VIP subtypes. Therefore, and as suggested by the reviewer, we have now toned down our statements pertaining to the molecular and cellular subtyping of the spiny VIP neurons.

To substantiate the electrophysiological characterization, we have added Supplementary Figure 7, which now better illustrates the difference in the spiking patterns between non-spiny and spiny neurons. The bursting phenotype corresponds with the bursting characteristics that have been described previously, i.e. the BS cells in Prönneke et al. *Cerebral Cortex* 2019 and the fAD and bNA cells in He et al. *Neuron* 2016. This aligns with the possibility that the spiny cell population could include those expressing CCK or CR.

Minor:

The properties of EPSCs have been evaluated in spiny and non-spiny neurons (Fig. 2). However, it is not clear how EPSCs were separated from IPSCs. The recordings were obtained at a holding potential of -70 mV, and the intrapipette solution contained 24 mM Cl⁻, leading to the reversal potential for Cl⁻ around -50 mV. This configuration predicts that spontaneous PSCs recorded at -70 mV should be a mixture of EPSCs and IPSCs, as a portion of inward currents may originate from opening of GABA-A receptors.

We thank the reviewer for pointing this out. It made us realize that we had given a wrong composition of the external and internal solutions. We apologize for the mistake. We now list the correct external and internal solutions.

The extracellular solution used during the recordings contained (in mM): 124 NaCl, 3 KCl, 2 CaCl₂, 1.3 MgSO₄, 26 NaHCO₃, 1.25 NaH₂PO₄, 10 D-glucose with an osmolarity of 300 mOsm and pH: 7.4, bubbled with 95% O₂-5% CO₂. The internal solution contained (in mM) 110 K-gluconate, 10 KCl, 10 HEPES, 4 ATP, 0.3 GTP, 10 phosphocreatine and 0.4% biocytin and pH between 7.2 and 7.3. With this combination, the reversal potential for chloride is around -68 mV. With a holding potential of -70 mV we could reliably detect and measure spontaneous EPSCs.

The Authors claim in the text that 80% of synapses on VIP dendrites were excitatory, a ratio that fits to the general ratio of excitatory and inhibitory neurons in cortical structures. However, earlier EM studies showed that inputs on interneurons expressing CR (and likely VIP) and CCK, receive a disproportionately higher number of inhibitory synapses as the general ratio of excitatory and inhibitory cells predicts (Gulyas et al., 1999; Matyas et al., 2004). This discrepancy may deserve discussion.

We thank the reviewer to pointing us to these two papers. Indeed, the fractions that we find are lower than what has been described in these two papers. However, it is important to note that these studies were on various inhibitory cell types in the hippocampus, whereas ours specifically pertain spiny VIP dendrites in L1 of cortex. Therefore, distinctions in the location and cell subtypes may explain these differences. Nonetheless, we have included a few lines on this difference in the discussion and incorporated the references.

Reviewer #2 (Remarks to the Author):

I am satisfied with the changes the authors have made to the manuscript. This is a great study!

We thank the reviewer for her/his time and efforts in reading and commenting on the manuscript.

REVIEWERS' COMMENTS:

Reviewer #1 (Remarks to the Author):

The MS has been substantially improved after the revision. I have only one comment. In the Introduction, the Authors may consider citing the papers that first described the presence of VIP-containing dis-inhibitory circuits in the hippocampal networks (e.g. Acsady et al., 1996). The seminal discovery of GABAergic cells that selectively innervate other GABAergic cells by Tamas Freund's group introduced a new functional unit in cortical circuits, a finding that may worth mentioning in a publication focusing on cortical VIP interneurons.